# A Systematic Review of Biosynthesized Metallic Nanoparticles as a Promising Anti-Cancer-Strategy

**DOI:** 10.3390/cancers13112818

**Published:** 2021-06-05

**Authors:** Anisa Andleeb, Aneeta Andleeb, Salman Asghar, Gouhar Zaman, Muhammad Tariq, Azra Mehmood, Muhammad Nadeem, Christophe Hano, Jose M. Lorenzo, Bilal Haider Abbasi

**Affiliations:** 1Plant Cell and Tissue Culture Lab, Department of Biotechnology, Quaid-i-Azam University, Islamabad 45320, Pakistan; ansaandleeb097@gmail.com (A.A.); gzaman@bs.qau.edu.pk (G.Z.); 2Proteomics Lab, School of Biochemistry & Biotechnology, University of the Punjab, Lahore 54590, Pakistan; aneetaandleeb3@gmail.com; 3Media and Production Group, Centre for Media and Communication Studies, University of Gujrat, Gujrat 50700, Pakistan; salmanasghar97@gmail.com; 4Nanobiotechnology Group, Department of Biotechnology, Mirpur University of Science and Technology, Mirpur 10250, Pakistan; tariq.awan@must.edu.pk; 5Stem Cell & Regenerative Medicine Lab, National Centre of Excellence in Molecular Biology, University of Punjab, 87-West Canal Bank Road, Lahore 53700, Pakistan; azramehmood@cemb.edu.pk; 6Department of Biotechnology, Institute of Integrative Biosciences, Peshawar 25100, Pakistan; m.nadeem@cecos.edu.pk; 7Laboratoire de Biologie des Ligneux et des Grandes Cultures (LBLGC), INRA USC1328 Université ď Orléans, CEDEX 2, 45067 Orléans, France; hano@univ-orleans.fr; 8Centro Tecnológico de la Carne de Galicia, Avd. Galicia no 4, Parque Tecnológico de Galicia, San Cibrao das Viñas, 32900 Ourense, Spain; jmlorenzo@ceteca.net; 9Área de Tecnología de los Alimentos, Facultad de Ciencias de Ourense, Universidad de Vigo, 32004 Ourense, Spain

**Keywords:** cancer, cancer development, green synthesis, metallic NPs, anti-cancer effect

## Abstract

**Simple Summary:**

Cancer is one of the major public health burdens in the world. To date, various conventional cancer therapies have been used, but these therapies are less effective and have severe side effects. Currently, in order to find a better cure for cancer, researchers have tried to explore new approaches with minimal toxicity and fewer side effects. In recent years, nanotechnology has been widely used in diseases management and holds a promising future in curing complex incurable diseases, in particular cancer. Biosynthesized metallic nanoparticles are eco-friendly and biocompatible, and can be used in cancer diagnostics, novel treatments, and drug delivery systems. This review gives an overview of the recent advancements in the biosynthesis of metallic nanoparticles (silver (Ag), gold (Au), zinc (Zn) and copper (Cu)) and their possible anti-cancer activities, with particular emphasis on the mechanisms of action, and future research prospects of nano-therapeutics are also discussed.

**Abstract:**

Cancer is one of the foremost causes of death worldwide. Cancer develops because of mutation in genes that regulate normal cell cycle and cell division, thereby resulting in uncontrolled division and proliferation of cells. Various drugs have been used to treat cancer thus far; however, conventional chemotherapeutic drugs have lower bioavailability, rapid renal clearance, unequal delivery, and severe side effects. In the recent years, nanotechnology has flourished rapidly and has a multitude of applications in the biomedical field. Bio-mediated nanoparticles (NPs) are cost effective, safe, and biocompatible and have got substantial attention from researchers around the globe. Due to their safe profile and fewer side effects, these nanoscale materials offer a promising cure for cancer. Currently, various metallic NPs have been designed to cure or diagnose cancer; among these, silver (Ag), gold (Au), zinc (Zn) and copper (Cu) are the leading anti-cancer NPs. The anticancer potential of these NPs is attributed to the production of reactive oxygen species (ROS) in cellular compartments that eventually leads to activation of autophagic, apoptotic and necrotic death pathways. In this review, we summarized the recent advancements in the biosynthesis of Ag, Au, Zn and Cu NPs with emphasis on their mechanism of action. Moreover, nanotoxicity, as well as the future prospects and opportunities of nano-therapeutics, are also highlighted.

## 1. Cancer: A Global Public Health Issue

Cancer is one of the leading causes of death, resulting in about 10.0 million deaths in 2020 alone [1]. Additionally, according to the World Health Organization (WHO), it is anticipated that it will increase up to three folds by the end of 2040 [2,3]. Cancer causes one in six deaths globally, resulting in more deaths than tuberculosis, malaria and acquired immunodeficiency syndrome (AIDS) [4]. Around 70% of these deaths occur in low- and middle-income countries owing to their lifestyle adaptations [3]. Chemotherapy, surgery, radiations, immunotherapy, and hormone therapy are commonly used for cancer treatment, but these approaches pose severe side effects in patients [5,6]. Chemotherapeutic agents cause various toxicities, for example, a commonly used drug, 5-fluorouracil, is generally associated with myelotoxicity, leukopenia, cardiotoxicity, and blood vessels constriction [7]. Similarly, cyclophosphamide and bleomycin, often used in combination therapy, are associated with bladder toxicity, pulmonary toxicity, and cutaneous toxicity [7,8,9]. Doxorubicin, another anticancer drug, is reported for cardiotoxicity, myelotoxicity, and renal toxicity, respectively [10]. In order to find a better cure with minimal toxicity, scientists are on a quest to explore novel approaches and discover potent anticancer agents for effective treatment against cancer with minimal side effects.

In the recent years, nanotechnology based therapeutic and diagnostic approaches have shown significant potential to ameliorate cancer therapy [3,11]. Cancer nanotechnology developed a new area of integrative research in biology, chemistry, engineering, and medicine, and is concerned with major advances in cancer diagnosis, prevention and treatment [12]. In past few years, nanoparticles (NPs) have become a subject of attraction for scientists due to their maximal efficacy and safety [13]. Due to these applications, recently, the US FDA has approved nanotechnology based anticancer drugs such as, Myocet™ (Perrigo, Dublin, Ireland), DaunoXome^®^ (Gilead Sciences, Foster City, CA, USA), Doxil^®^ (Johnson & Johnson, New Brunswick, NJ, USA) and Abraxane^®^ (Celgene, Summit, NJ, USA) [14].

This article provides an insight into the green synthesis of metallic NPs and their potential applications as therapeutics in cancer therapy. This review has mainly focused on biosynthesis of silver, gold, zinc and copper NPs for cancer therapy and their in vitro anticancer activities against cell lines. The basic mechanism behind cancer development and a proposed mechanism involved in metallic NPs-mediated cytotoxicity in cancerous cells have also been discussed in the current review.

## 2. Genome Instability: A Basic Mechanism in Cancer Development

The fundamental abnormality that leads to the development of cancer is the abnormal cellular proliferation and division, which arise when their regulatory genes are mutated [15]. The protein product of these mutated genes can cause cancer by accelerated cell division rates or inhibiting normal cell cycle control, such as programmed cell death or cell cycle arrest [16]. The genes that mainly contributed to development of cancer fall into three broad categories, involving proto-oncogenes, oncogenes, and tumor suppressor genes. The proto-oncogenes (normal version of genes), when activated or mutated, become oncogenes (mutated version of genes) and produce various onco-proteins that can affect cell division, proliferation and survival, and results in cancer development [17,18]. A few of the many known proto-oncogenes include *HER-2/neu, RAS, MYC, SRC, BCL-2* and *hTERT*, and these genes or their product modulate cellular cycle or control normal cell division or apoptosis cell division [19,20,21,22,23,24]. On the contrary, tumor suppressor genes code proteins that repair damaged DNA or destroy damaged cells, and when these molecular switches become mutated, it leads to abnormal cell division and cellular growth. In this way, the abnormal cells continue to survive and may eventually develop into a cancer [25].

Cancer is mainly associated with loss of genome stability. Genome stability of cells is mostly altered through certain DNA damaging agents from carcinogens. Fortunately, our cells have proofreading machinery such as cell cycle checkpoints and a complex interconnected network of pathways to repair the damage [26]. However, mutation can occur in the regulatory genes and the cell will be unable to proofread such DNA breakages, and eventually the normal cellular cycle and proliferation rate will be disrupted [27,28]. For example, *Rad54B* is an important protein that exhibits a role in DNA repair and maintaining genome stability after DNA damage [27]. Various studies [29,30,31] have revealed that *Rad54B* mutation is involved in the development of some cancer’s cells, and such abnormal proteins are unable to terminate the cell cycle and will lead to the progression of cancer. Figure 1 shows regulation of cell cycle upon DNA damage and the role of *Rad54B* in the development of cancer.

The development of cancer is preceded by the appearance of mutations in critical cellular genes involved in regulatory pathways of the cell cycle. This is the initial stage (initiation) of cancer development, an irreversible heritable alteration in DNA of normal cell referred to as initiated cell [32]. Initiation is associated with high efficacy of DNA repair, otherwise the initiated cell may ultimately die while progressing towards the development of the preneoplastic focal lesions. The initiated cells in preneoplastic focal lesions starts proliferating upon continual exposure to promoting agents, and further mutations during promotion leads to development of metastasis or neoplasm [33]. Neoplasia, an abnormal or uncontrolled growth of cells or tissues, can be benign (localized tumor) or malignant, which tend to proliferate rapidly, or metastasize (spread the tissues around them or other parts of the body) [34]. Figure 2 shows different stages of development of cancer; starting from a mutation in normal cells (initiation), proliferation of mutated cells (promotion), and uncontrolled growth of cells along with continued mutations in their genome, and their spread to other parts of the body (metastasis).

## 3. Green Synthesized Metallic NPs: An Insight

Several metals and their oxides have been used for production of NPs, including silver (Ag), aluminum (Al), iron (Fe), gold (Au), silica (Si), copper (Cu), zinc (Zn), manganese (Mn), cerium (Ce), titanium (Ti), platinum (Pt) or thallium (TI) [35]. NPs are generally synthesized by via approaches, top-down approach and bottom-up approach, as shown in Figure 3. The top-down approach for NPs synthesis includes lithographic techniques, laser ablation, ball mining, sputtering, electro-explosion and etching. The bottom-up approach includes the most effective methods for NPs synthesis, where NPs are prepared using simpler molecules [36].

From all the approaches of NPs synthesis, green synthesis approach is considered the most economic, sustainable, reliable and eco-friendly [37]. This approach of NPs synthesis does not require toxic chemicals, high temperature, high pressure and does not cause harm to human health and the environment [38]. At present, it is also considered a preferred method for NPs fabrication because of utilization of low-cost and non-hazardous raw material such as microorganisms fungi [39], algae [40], bacteria [41], plant extracts [42], natural polymers and proteins [43]. These resources contain biomolecules such as proteins including enzymes, polysaccharides, sugars, amides, ketones, aldehydes, and carboxylic acids, but also more importantly various phytochemicals such as terpenes, alkaloids or polyphenols including flavonoids that aid in immediate reduction (Figure 4).

For the reduction of metal ions, bacteria and fungi require a relatively extended incubation period compared to water-soluble phytochemicals that do it immediately in a much lesser time. Moreover, plants are considered better candidates for NPs synthesis as compared to microbes such as fungi and bacteria because, in case of plants, the intricate process of maintaining microbial cultures is eliminated. Another striking feature of biological synthesized NPs is their biocompatible nature. In contrast, the chemical route uses toxic reducing agents, thus limiting their biomedical potentials, and posing a threat to the ecosystem. Biological approach resolves this issue by using safe reducing agents and could be used in cancer therapeutics [44,45].

## 4. NPs for Cancer Therapy

At present, there are several treatment approaches are available for cancer, including radiation therapy, chemotherapy, immunotherapy, photodynamic therapy, cancer vaccinations, stem cell therapy and surgery, but these treatment options cause severe side effects and have pharmacokinetics issues [46,47]. NPs are progressing as an attractive tool of research to overcome these challenges [48]. NPs exhibit large surface to volume ratio, which is responsible for their interaction with the biological system because at the cell level, the atoms are freely available to commence various reactions [49,50]. These unique morphologies of NPs effect their insertion or entry into the cells. The charge present on the surface of NPs affects their circulation time in the blood stream and their rate of uptake and translocation. Cationic NPs apparently damage plasma-membrane integrity, hampers organelles architecture, and imbalance the normal cellular function compared to anionic NPs [51]. Hence, in this way, cationic NPs often show a higher rate of non-specific uptake as compared to neutral and negatively charged NPs. However, the neutral and negatively charged NPs exhibit shorter blood circulation time, which reduces their bioavailability [50]. It has been reported previously that positive groups like primary amine present at the surface of polystyrene microparticles helped in faster internalization in cells as compared to the microparticles, which contained hydroxyl, sulfate or carboxyl as surface groups [52]. Additionally, mesoporous silica NPs containing amine groups were used earlier in in vitro and in vivo studies as gene delivery tools and exhibited improved internalization owing to the positive groups on their surface [53].

NPs are attracting significant interest as carriers for diagnostic, hydrophobic medicine, hyperthermia, therapeutics and especially in delivery of antineoplastic drugs/agents to the cancerous tissues, where the delivered NPs can penetrate deep and deliver drug to a specific targeted site [54]. In cancerous cells, NPs have been reported to increase the intracellular concentration of drugs via either active targeting or passive targeting by minimizing toxicity to the normal cells [55]. Moreover, as a targeted drug delivery system, NPs have been developed as temperature- or pH-sensitive carriers. As a temperature-sensitive drug delivery system, these NPs can deliver and release drugs in the tumor area, by undergoing local changes in temperature via providing ultrasound waves or magnetic fields. The pH-sensitive system can carry and release drugs efficiently in the acidic environment of the cancerous cells [56]. These NPs can be further modified with specific targeting moieties, such as antibody fragments, antibodies, specific molecules, RNA aptamers and small peptides, which further enhance their ability to selectively bind to cancerous cells and tissues [57].

Angiogenesis (formation of new blood vessels) plays a key role in progression of a tumor towards metastasis. Cancer cells display abnormal membrane structure because of enhanced blood vasculature due to upregulated expression of angiogenic factors [58,59]. This dysregulated membrane architecture, can be of great interest to deliver anti-angiogenic nano-based targets into the tumor microenvironment to inhibit excess production of angiogenic stimulators [60]. Owing to effectiveness of this therapy, several studies have been reported to block signaling of *VEGF, PDGF, EDGR,* angiopoietin- key contributors of neovascularization [61]. Nano anti-angiogenic therapy can be a good delivery option for drugs that have a short half-life, poor oral availability, and distribution in tumor area [59]. Depending on their sizes, NPs can easily penetrate the tumor microenvironment and can efficiently deliver antiangiogenic drugs. Through enhanced permeability and retention effect (EPR), the NPs with optimum size can intrinsically approach the metastasized tumors and can efficiently release loaded drugs as shown in Figure 5 [60].

## 5. The Fate of Cancer Cells Exposed to NPs

Metallic NPs offer more cytotoxicity to cancerous cell lines as compared to normal cells [62,63]. Various mechanisms have been proposed to explain the cytotoxicity mechanism of metallic NPs such as generation of reactive oxygen species (ROS), activation of caspase-3, permeabilization of mitochondrial outer membrane, and specific DNA cleavage, all of which lead to apoptotic, autophagic and necrotic death of the cancer cell [64]. Figure 6 demonstrates an overview of the proposed cytotoxicity mechanism of metallic NPs against cancerous cells.

NPs of different sizes (either small or large) follow different mechanisms to enter the cells. Smaller NPs get into the cells via receptor-mediated uptake by developing interactions with the caveolin receptor present on the cell membrane. Larger NPs tend to enter the cells via clathrin-mediated endocytosis. Once they make entry to the cells NPs take different paths within the cell to perform their directed function, either they directly interact with the proteins in cytosol, or they undergo some surface modifications in the lysosome–endosome complex before release into the cytosol [64]. Inside the cell, NPs trigger a cascade of ROS and start releasing metal ions, which tend to bind with the SH groups of proteins and results in breakage of its S–S bridges. In this way, the physiology of the cell is affected, resulting in activation of several signaling pathways that leads to programmed cell death [65].

Apoptosis is often triggered either by intrinsic or by extrinsic pathways. Nanomaterials can activate apoptotic signaling by both intrinsic and extrinsic pathways. In case of apoptosis triggered via intrinsic pathway, ROS generation results in mitochondrial membrane depolarization, which leads to release of cytochrome *c* into the cytosol. This cytochrome *c* then leads to activation of caspase-9/3 apoptotic cascade by triggering pro-apoptotic proteases in apoptosis initiated by extrinsic pathway [66].

Autophagy is also a form of programmed cell death and is well controlled by autophagy-related genes (ATGs). Autophagy is stimulated by extracellular or intracellular stress, that is generally cytoprotective in nature and leads to cell survival, whereas an over-stimulation of autophagy causes cytotoxicity and may lead to autophagic cell death [66]. Nanomaterials can initiate autophagy through various pathways such as, aggregation of impaired proteins, which can cause organelle stress, oxidative stress, variation in gene expression, and inhibition of kinase-mediated regulatory pathways [67]. The elevated level of autophagic vacuoles in the cells as a response to nanomaterials could be a type of adaptive cellular response. Previous studies showed that nanomaterials can generate elevated levels of autophagic vacuoles as noticed in in vitro studies conducted on various animals and human cells and in in vivo models [68]. Before entering the cytoplasm, silver NPs undergo degradation within a double-membraned autophagosome compartment [69].

Programmed necrosis is also termed as programmed cell death, which involves binding of death ligands to their receptors. The ligation to death receptor leads to a complex formation, and this pro-necrotic complex further binds with metabolic enzymes and results in increased ROS production, which activates necrosis. Nanomaterials can induce ROS-mediated necrosis directly by affecting mitochondria or indirectly by elevating NADPH oxidase and cellular calcium levels, to generate more ROS and undergo programmed necrosis [65].

Subcellular location of NPs also plays an important role in death of cancer cells. NPs took 30–60 min for their release from the endosome, while NPs that are aggregated in multi-vesicular bodies are removed within a period of 6 days. Similarly, Golgi apparatuses also extruded the particles assembled in the microtubule [70].

## 6. Anti-Cancer Activities of Biosynthesized Metallic NPs

There are various advantages of using plants for NPs synthesis, because they are safe to handle, are easily available and contain a vast variety of biomolecules or metabolites that help in stabilization and reduction of NPs [71].

In modern medicine, plant-based nanotherapeutics drugs have become a potential weapon in cancer therapeutics. In recent years, optimal methods for metallic NPs preparations with anti-cancer properties are widely being examined both in vivo and in vitro [72]. Plant extract and bioactive compounds of several medicinal plants have been reported for their potential use as anticancer agents [73]. The mechanism of action of against cancer have been extensively studied by researchers and found that the functional groups capped on the NPs are involved directly or indirectly in improving the anticancer activity or reducing the toxicity or improving the bioavailability and uptake [74]. The anticancer properties of different NPs also exhibit variations because of differences in phytocontent of biological material used for their synthesis [75]. Figure 7 presents a schematic representation of synthesis of plant-based metallic NPs and their application as anti-cancer therapeutics. Here, it is worth mentioning that in order to keep this review article less verbose we have only discussed plant based green synthesis of silver (Ag), gold (Au), zinc (Zn) and copper (Co) metals and their inhibitory activities against several cancerous cell lines.

### 6.1. Applications of Biosynthesized Silver NPs (AgNPs) as Anti-Cancer Therapeutics

Among all the noble metals, silver has received major attention from researchers due to its unique surface chemistry and morphologies [76].

According to the literature, biosynthesized AgNPs have displayed significant anticancer potential against the cervical cancer cell lines HeLa and Siha. Hexagonal and triangular shaped AgNPs sizes ranging from 2–18 nm have shown notable inhibitory actions against Siha cancer cell line with an ≤4.25 μg/mL IC_50_ value [77]. In contrast, growth of the HeLa cancer cell lines was successfully inhibited by AgNPs, which are spherical in shape with sizes ranging from 5–120 nm. NPs preparation from different plants exhibited a diverse range of IC_50_ values that depended on the method used for AgNPs synthesis and the type of plant extracts used [78,79,80,81,82,83,84,85,86]. The spherically shaped bio-synthesized AgNPs with sizes ranged between 7.39–80 nm have displayed inhibitory activities against colon cancer cell lines HCT 15, HT29 cells and HCT-116, and their IC_50_ values ranged between 5.5–100 μg/mL [87,88,89,90,91]. Inhibition by biosynthesized AgNPs have been successfully carried out against lung cancer cell line A549. The prepared NPs were spherical in shape with sizes ranging between 13–136 nm and showed a dose-dependent inhibitory activity with different values of IC_50_ and LD_50_, as mentioned in Table 1 [77,92,93,94,95,96,97,98,99]. Spherical biosynthesized AgNPs with sizes ranged between 5–50 nm inhibited the human gastric adenocarcinoma (AGS) cell line with 21.05 μg/mL IC_50_ value [100].

Inhibition by spherical biosynthesized AgNPs with sizes ranged between 6.4–27.2 nm was observed against the intestinal cancer cell line SMMC-7721 with above 27.75 μg/mL IC_50_ value [101]. Bio-extract-derived AgNPs, which are spherical and cuboidal in shape with sizes ranged between 59–94 nm, showed inhibitory actions against epidermoid carcinoma cell line A431, where IC_50_ values ranged between 78.58–83.57 μg/mL [102]. Many biosynthesized AgNPs were reported that inhibited the growth of the MCF-7 breast cancer cell lines or showed toxicity against them. The shapes of AgNPs reported in these anticancer studies varied such as cuboidal, hexagonal, spherical, and pentagonal with the sizes ranging between 5–80 nm and their IC_50_ values ranging between 3.04–250 μg/mL. Additionally, some studies outcomes suggested that the IC_50_ values of biosynthesized AgNPs varied in a dose dependent fashion and depended on the dose of the extract used [82,89,96,98,103,104,105,106,107,108,109,110,111,112]. Spherical AgNPs that ranged from 31–56 nm in size repressed the laryngeal carcinoma cell line Hep-2, with IC_50_ values ranged between 3.42–12.5 μg/mL [109,113,114]. The hepatic cancer cell lines Hep-G2 were inhibited by spherically shaped AgNPs [115,116]. Inhibition by biosynthesized spherical AgNPs that were 8–22 nm in sizes was observed against leukemia cell lines HL-60 and H1299 with 5.33 μg/mL IC_50_ value, and the inhibition depended on the dose of extract used for preparation [79,117]. The kidney cancer cell line Hek-293 was inhibited by 40 nm spherical AgNPs in a dose-dependent fashion [82]. Many studies have been conducted for biosynthesis of AgNPs and to ascertain their impact on various cancerous cell lines. AgNPs are known to possess anti-angiogenic properties. In one of the studies, performed on bovine retinal epithelial cells (BRECs), AgNPs of 40 nm size were shown to successfully reduce VEGF-induced angiogenesis by inhibiting the PI3K/Akt signaling pathway [3].

The available data related to biosynthesized AgNPs against cervical cancer, colon cancer, lung cancer, gastric carcinoma, intestinal cancer, epidermoid carcinoma, breast cancer, hepatic cancer, laryngeal carcinoma, leukemia, and kidney cancer are enumerated in Table 1.

### 6.2. Applications of Biosynthesized Gold NPs (AuNPs) as Anti-Cancer Therapeutics

Besides silver, gold is also considered a good candidate for NPs synthesis, showing high dispersion owing to their small size and large surface area. Moreover, due to its resistance to oxidation by moisture, air and acids and biocompatible nature, it has gained attention in the biomedical field, particularly in areas of cell targeting, tumors detection, drug-delivery and cancer therapy [118]. It was reported recently that AuNPs are more effective in drug delivery due to their self-assembled natural [119] and for hyperthermia because of their optical excitation properties [120].

A series of in vitro studies has been conducted on various cancer cell lines, to evaluate the anticancer potential of biosynthesized AuNPs. Biosynthesized AuNPs, which are spherical in shape with sizes ranged between 12–30 nm, showed inhibitory actions against MCF-7 breast cancer cell lines and their IC_50_ values depended on the method used for AuNPs synthesis and the type of plant extracts used for their preparation [121,122]. Spherical and triangular shaped AuNPs of sizes ranged between 13–28 nm showed cytotoxicity against MCF-7 breast cancer cells with a 257.8 µg/mL IC_50_ value [123].

In other studies, AuNPs which were spherical in shape with sizes ranged between 22–30 nm showed cytotoxicity against MDA- MB-231 breast cancer cell lines by activating apoptotic cell death pathways [124]. Bio-extract derived AuNPs with 14.6 nm size exhibited inhibitory actions against breast cancer cells through DNA damage and necrosis [125]. Spherically shaped biosynthesized AuNPs with average sizes of 95 nm repressed the growth of breast cancer cells MCF-7 by regulating the expression of anti-apoptotic (p53) and pro (Bcl-2) proteins with a 4.76 μg/mL IC_50_ value [126]. In another study, inhibition of breast cancer cell line HBL-100 was shown by spherically shaped AuNPs [127]. Inhibition by biosynthesized AuNPs that exhibited spherical and aggregated morphology was observed against A549 lung cancer cell lines. The size of these AuNPs ranged between 80–120 nm and offered cytotoxicity to cancerous cell lines by up-regulating many proinflammatory genes such as tumor necrotic factor-alpha (TNF-α) and interleukins IL-10 and IL-6 [128]. Inhibition of A549 lung cancer cell lines was shown by AuNPs, which are spherical in shape with 14 μg/mL IC_50_ value [129].

Biosynthesized AuNPs, which were hexagonal, triangular, and quasi-spherical in shape with sizes ranged between 6.03–150 nm repressed the A549 Lung cancer cell lines by offering low toxicity [130,131]. Pentagonal and triangular shaped biosynthesized AuNPs with sizes ranged between 10–50 nm showed substantial anticancer potential against cervical cancer cell lines HeLa by inhibiting their proliferation with an IC_50_ value of 100 µg/mL [132]. Other studies against cervical cancer using HeLa cell lines demonstrated the inhibitory activities of biosynthesized AuNPs derived from various plant extracts, their sizes and IC_50_ values varied and were dependent upon the type and dose of respective plant extracts used [133,134,135,136]. Moreover, cytotoxicity testing of biosynthesized AuNPs has been conducted on various other cell lines such as kidney [122,137,138], leukemia [139], and liver [140], as mentioned in Table 2.

Available data regarding anticancer activities of biosynthesized AuNPs against the cell lines mentioned above for cervical cancer, breast cancer, lung cancer, kidney cancer, leukemia and liver cancer are summarized in Table 2 with their citations.

### 6.3. Applications of Biosynthesized Zinc and Zinc Oxide NPs (Zn/ZnO-NPs) as Anti-Cancer Therapeutics

Biological synthesis of zinc and zinc oxide NPs are of great interest in recent years for the fabrication of eco-friendly NPs because of presence of phytochemical components like flavonoids, phenolics or alkaloids [141]. The specific physicochemical properties of ZnO NPs helps in their cellular uptake and their innate toxicity against cancerous cells can induce intracellular ROS generation, which ultimately leads to death via an apoptotic pathway, these characteristics make them an attractive candidate for biomedical applications [142].

Different parts of the plants have been extensively studied for the biosynthesis of ZnO NPs and their anticancer effects have been investigated in vitro using various cancerous cell lines. Spherical and hexagonal shaped bio-extract-derived Zn NPs have shown cytotoxicity in lung cancer cell lines A549 and Calu-6. These NPs exhibited various sizes and IC_50_ values depending on the types of plant extracts used for their preparation and their doses used [142,143,144,145,146]. Spherical and hexagonal biosynthesized ZnNPs with sizes ranging between 22.5–50 nm, prepared from different plant extracts, inhibited the WEHI-3 leukemia cancer cell lines, with IC_50_ values ranging between 2.25–12.4 μg/mL [147,148]. Spherical biosynthesized ZnNPs of cell lines and their IC_50_ values varied in a dose dependent manner and dependent upon the type of plant extracts used [106,149,150,151,152,153,154,155]. Biosynthesized hexagonal ZnNPs with sizes 10 ± 1.5 nm showed inhibitory actions against CaOV-3 ovarian cancer cell lines with IC_50_ value of 10.8 ± 0.3 μg/mL [156]. Inhibition by biosynthesized spherical ZnNPs that were 47 nm in sizes was observed against colon cancer cell lines HT-29 with 9.5 μg/mL IC_50_ value, respectively [157]. Similarly, biosynthesized ZnO-NPs showed potential inhibitory activities against epidermoid carcinoma cell lines A43 with an IC_50_ value of 16.5 ± 1.6 μg/mL [158], and against liver cancer cell lines Hep-G2 with an IC_50_ value of 14.1 ± 0.7 μg/mL [159]. Table 3 explains anticancer activities of ZnO NPs against lung cancer, breast cancer, ovarian cancer, colon cancer, epidermoid carcinoma, and liver cancer cell lines.

### 6.4. Applications of Biosynthesized Copper/Copper Oxide NPs (Cu/CuO-NPs) as Anti-Cancer Therapeutics

Copper NPs have also gained significant attention as cytotoxic nano-entities because of their low cost, easy availability, and great similarity in properties with the noble metals [161]. Copper and copper oxide NPs are extensively used as a tool for cancer imaging owing to their highly effective light-to-heat transformation property under influence of near-infrared laser irradiation [162].

Different biologically synthesized Cu/CuO NP have been shown to be cytotoxic against multiple cancerous cell lines. Plant-mediated biosynthesized CuO NPs, which were spherical and hexagonal in shape with sizes of 26.6 nm exhibited inhibitory actions against cervical cancer cell lines HeLa by initiating ROS mediated apoptotic pathways [163]. Similarly, spherically shaped CuO NPs of 12 nm sizes, prepared from aqueous leaf extracts of different plants showed cytotoxicity against cervical cancer cell lines HeLa, breast cancer cell lines MCF-7 and lung cancer cell lines A549, and their IC_50_ values varied depending on the types of plants used [164]. Inhibition of MCF-7 breast cancer cell lines were carried out using biosynthesized spherically shaped CuO NPs of 26–30 nm sizes with a 56.16 μg/mL IC_50_ value [165].

In another study, aqueous leave extract derived CuO NPs, which are spherical in shape with sizes ranging between 20–50 nm, showed the highest anticancer activity against AMJ-13 breast cancer cell lines with an IC_50_ value of 1.47 μg/mL and against SKOV-3 ovarian cancer cell lines with a 2.27 μg/mL IC_50_ value [166]. Biosynthesized CuO NPs with 577 nm sizes displayed cytotoxicity against lung cancer cell lines A549 through apoptosis initiated via nuclear fragmentation and showed an IC_50_ value of 200 μg/mL [167]. Similarly, spherically shaped CuO NPs of different sizes were tested against cervical cancer cell lines HeLa and lung cancer cell lines A549 [168]. Cytotoxicity of spherically shaped biosynthesized CuO NPs with about 4.8 nm sizes were tested against prostate cancer cell lines PC-3 [169]. Table 4 shows anticancer activities of Cu/CuO NPs against cervical cancer, breast cancer, ovarian cancer, lung cancer, and prostate cancer cell lines.

## 7. Nano-Toxicity, the Concern/Bottleneck

Despite their promising potential in biomedical field. there are certain adverse health effects linked with their use [170]. For instance, agglomeration is one of the leading problems in translating this therapy into medicines as it poses toxicity in organ systems. Even if not agglomerated, it causes cellular injuries [171]. Toxicity offered by NPs is generally attributed to their morphology and surface reactivity. The toxicity associated with NPs can be controlled by including free groups at their surfaces such as –COOH groups, which are considered less toxic than –OH group and –NH_2_ groups [172]. Toxicity can also be minimized by controlling the size (30–100 nm) of metal NPs [173]. For specific and targeted use of nanomaterials, it is essential to understand the possible interactions between biological systems and the NPs, in this way the aggressive reactions can be minimized. To reduce toxicity, biological synthesis of NPs is preferred due to occurrence of biocompatible phytoconstituents [174,175]. Some studies also indicated that polyphenol compounds are nontoxic to healthy cells while exhibiting toxicity against cancerous cells [176].

## 8. Conclusions and Future Prospects

Despite all the recent advancements in cancer diagnosis and treatment, cancer remains one of the main causes of death globally. To date, no efficient treatment has been discovered to treat cancer, and all of the available anticancer drugs hold potential side effects. Thus, in a quest to find better diagnostics and treatment with maximal efficiency, specificity and lesser toxicity, scientists are looking to develop novel approaches. Recently, biological, or green, synthesis of NPs has gained significant attention in the biomedical field. Green synthesis is cost effective, less toxic and eco-friendly as compared to other methods of NPs formulation. The higher biocompatibility, lesser agglomeration rate, maximal clearance and lesser toxicity are the main aspects to be considered, this review article gives a compendious idea about the green synthesis of metallic NPs (Ag, Au, Zn/ZnO and Cu/CuO) and their mechanism of action and explored their therapeutic potential in vitro against various cancer cell lines. The effect of NPs varied from one type of cancer to the other, indicating that besides the specific properties of NPs, the cellular response is also important. ROS is an initiator molecule in autophagic, apoptotic and necroptotic death pathways and hence, it can be considered as the precursor component of cell death.

Metallic NPs showed remarkable promises in case of nano-based medical treatments, but their 3D tumor model studies and clinical trials remain unexplored. Therefore, their clinical trials are compulsory for leading the future direction regarding their applications. Currently, analysis into their dose, route of administration and biodegradability are the main hurdles that need to be tackled in the clinical trials.

## Figures and Tables

**Figure 1 cancers-13-02818-f001:**
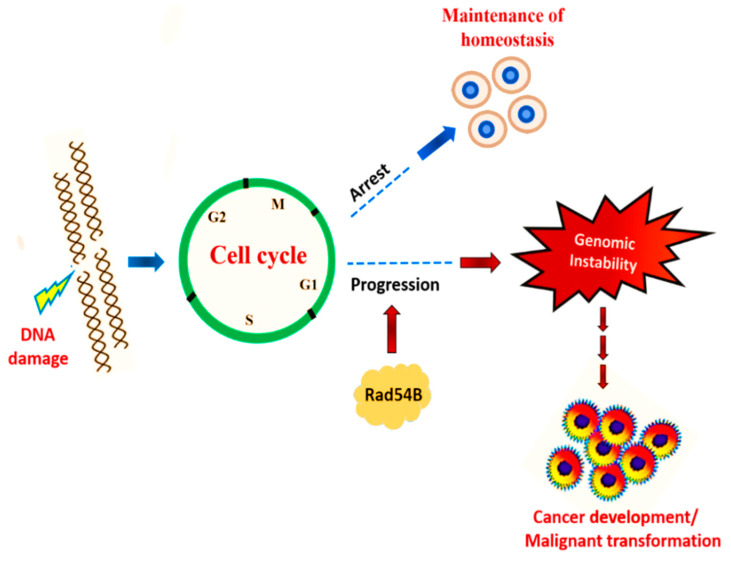
Cell cycle regulation in response to DNA damage.

**Figure 2 cancers-13-02818-f002:**
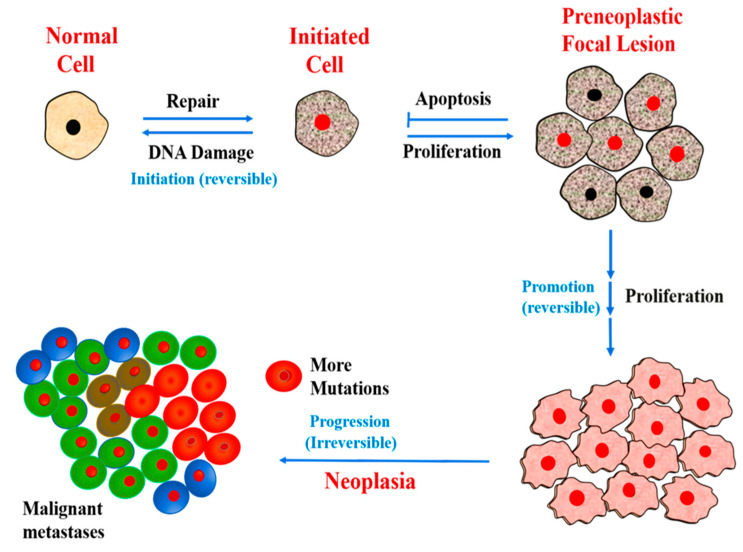
Stages in the development of cancer.

**Figure 3 cancers-13-02818-f003:**
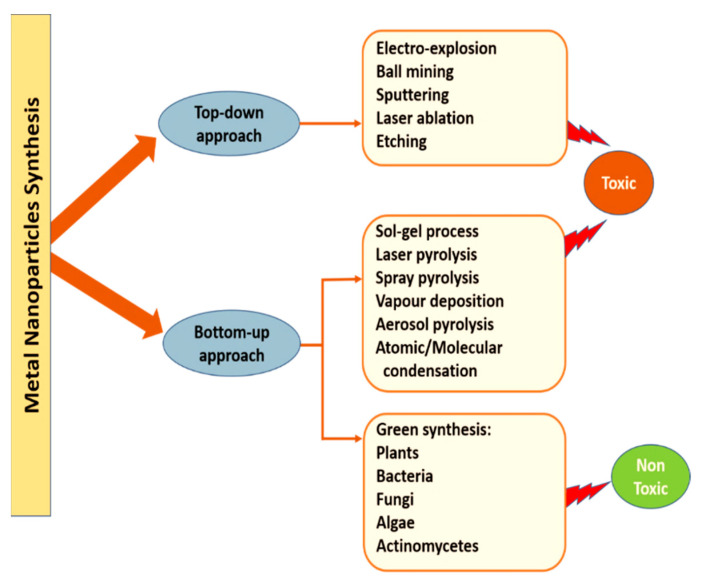
Different approaches of synthesis of metal NPs.

**Figure 4 cancers-13-02818-f004:**
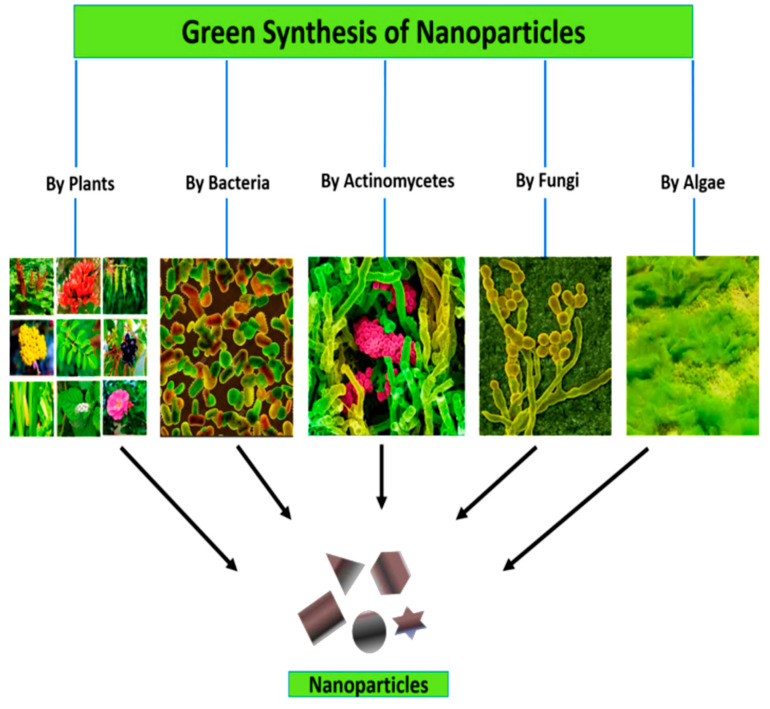
Green synthesis of NPs.

**Figure 5 cancers-13-02818-f005:**
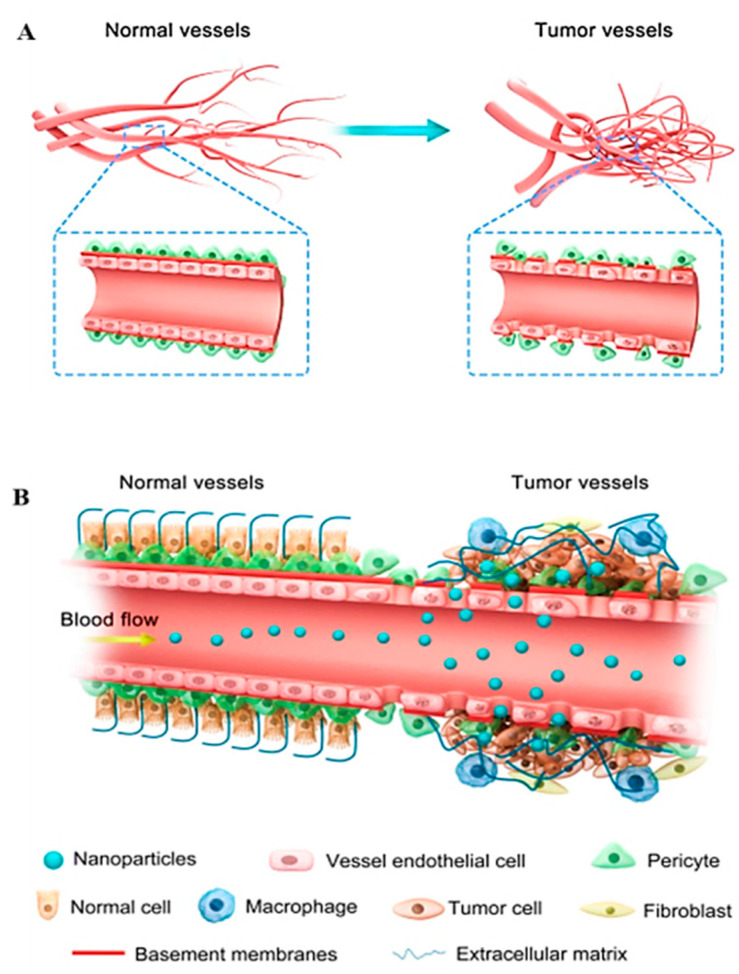
(**A**) illustrates the vasculature of both normal and tumor cells and (**B**) shows accumulation and penetration of NPs in tumor tissues via EPR effect. Reproduced from [60].

**Figure 6 cancers-13-02818-f006:**
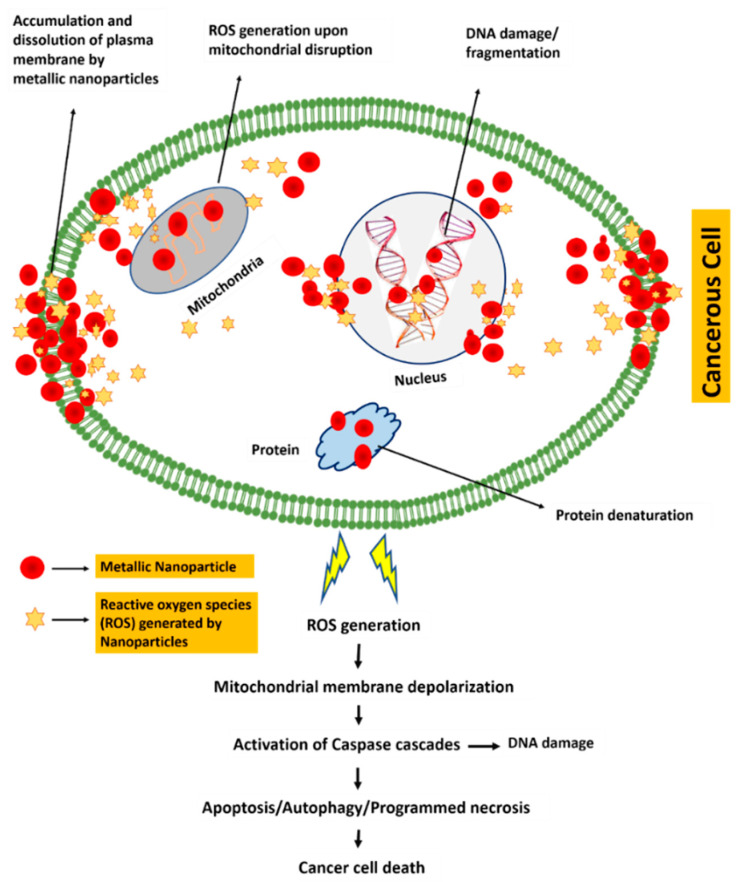
Proposed mechanism of green synthesized metallic NPs mediated cytotoxicity in cancerous cells.

**Figure 7 cancers-13-02818-f007:**
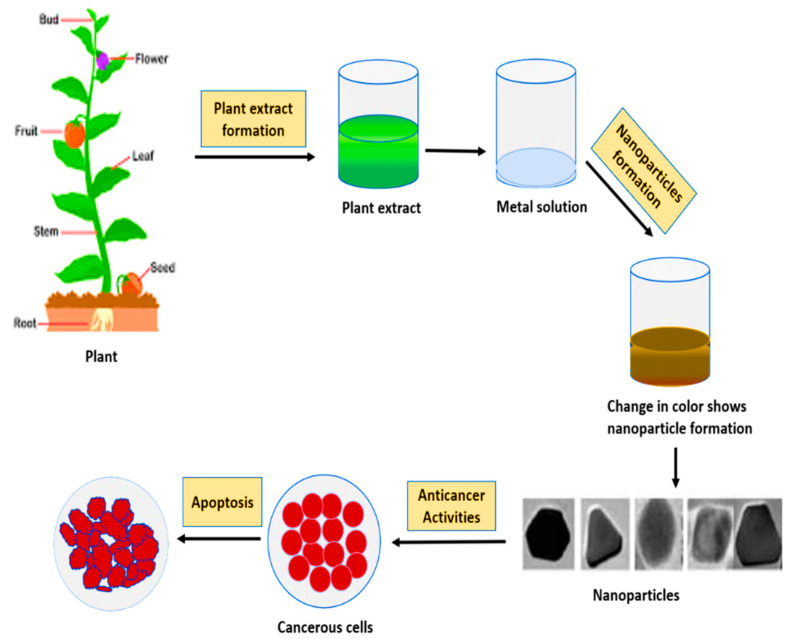
Plant mediated metallic NPs synthesis and their anticancer activity.

**Table 1 cancers-13-02818-t001:** List of studies exhibiting biosynthesized silver NPs and their anticancer activity.

Plant	Plant Part Used	Morphology/Size (nm)	Exposure Time	Cancer Type/Cell Line	IC_50_ Value	Ref.
*Moringa olifera*	Stem bark	Spherical/38–40	24 h	Cervical Cancer/HeLa	Dose dependent	[78]
*Sargassum vulgare*	Whole plant	Spherical/10	3 h	Cervical Cancer/HeLa	Dose dependent	[79]
*Melia azedarach*	Leaf	Spherical, cubical/78	10 min	Cervical Cancer/HeLa	300μg/mL (LD50)	[80]
*Podophyllum hexandrum*	Leaf	Spherical/14	30–150 min	Cervical Cancer/HeLa	20 μg/mL	[81]
*Syzygium cumini*	Leaf	Spherical/<40	6 h	Cervical Cancer/HeLa	Dose dependent	[82]
*Azadiracht a indica*	Leaf	Hexagonal, triangular/2–18	-	Cervical cancer/Siha	≤4.25 μg/mL	[77]
*Acorous calamus*	Rhizome	Spherical/31.86	20 h	Cervical cancer/Siha	Dose dependent	[83]
*Calotropis gigantea*	Latex	Spherical/5–30	24 h	Cervical cancer/Siha	Dose dependent	[84]
*Heliotropium indicum*	Leaf	Spherical/80–120	2 h	Cervical cancer/Siha	20 μg/mL	[85]
*Cymodocea serrulata*	Whole plant	Spherical/17–29	2 h	Cervical cancer/Siha	107.7 (GI50)	[86]
*Ulva lactuca (algae)*	Whole plant	Spherical/56	10 min	Colon Cancer/HT29	49 μg/mL	[87]
*Commelina nudiflora L.*	Whole plant	Spherical, triangular/24–80	24 h	Colon Cancer/HCT-116	100 μg/mL	[88]
*Citrullus colocynthis*	Leaf	Spherical/13.37	24 h	Colon Cancer/HCT-116	>30 μg/mL	[89]
*Citrullus colocynthis*	Seeds	Spherical/16.57	24 h	Colon Cancer/HCT-116	>30 μg/mL	[89]
*Citrullus colocynthis*	Fruit	Spherical/19.26	24 h	Colon Cancer/HCT-116	21.2 μg/mL	[89]
*Vitex negundo*	Leaf	Spherical/22	4 h	Colon Cancer/HCT 15	20 μg/mL	[90]
*Rosa indica*	Petal	Spherical/23.52–60.83	1 h	Colon Cancer/HCT 15	30 μg/mL	[91]
*Artemisia princeps*	Leaf	Spherical/20	15 min	Lung cancer/A549	Time dependent	[92]
*Gossypium hirsutum*	Leaf	Spherical/13–40	3 min	Lung cancer/A549	40 μg/mL	[93]
*Origanum vulgare*	Leaf	Spherical/136 ± 10.09	Temp. dependent	Lung cancer/A549	100 μg/mL (LD50)	[94]
*Rosa damascene*	Petal	Spherical/15–27	0–25 min	Lung cancer/A549	80 μg/mL	[95]
*Syzygium aromaticum*	Fruit	Spherical/5–20	20 min	Lung cancer/A549	70 μg/mL	[96]
*Acorous calamus*	Rhizome	Spherical/31.86	20 h	Lung cancer/A549	Dose dependent	[77]
*Cymodocea serrulate*	Leaf	Spherical/29.28	1 h	Lung cancer/A549	100 μg/mL (LD50)	[97]
*Olax scandens*	Leaf	Spherical/30–60	2 h	Lung cancer/A549	Dose dependent	[98]
*Scoparia dulcis*	Leaf	Spherical/15–25	1 h	Lung cancer/A549	Dose dependent	[99]
*Artemisia marschalliana*	Shoots	Spherical/5–50	5 min	Gastric cancer/AGS	21.05 μg/mL	[100]
*Taxus yunnanensis*	Callus	Spherical/6.4–27.2	10 min	Intestinal cancer/SMMC-7721	27.75 μg/mL	[101]
*Cucurbita maxima*	Petal	Spherical, cuboidal/76	5–60 min	Epidermoid cancer/A431	82.39 μg/mL	[102]
*Acorus calamus*	Rhizome	Spherical, cuboidal/59	5–60 min	Epidermoid cancer/A431	78.58 μg/mL	[102]
*Alternanthera sessilis*	Shoots/Aerial parts	Spherical/10–30	6 h	Breast cancer/MCF-7	3.04 μg/mL	[103]
*Andrographis echioides*	Leaf	Pentagonal, cubic, hexagonal/68.06	12 h	Breast cancer/MCF-7	31.5 μg/mL	[104]
*Butea monosperma*	Leaf	Spherical/20–80	2 h	Breast cancer/MCF-7	Dose dependent	[105]
*Citrullus colocynthis*	Roots	Spherical/7.39	24 h	Breast cancer/MCF-7	2.4 μg/mL	[89]
*Citrullus colocynthis*	Fruit	Spherical/19.26	24 h	Breast cancer/MCF-7	>30 μg/mL	[89]
*Citrullus colocynthis*	Leaf	Spherical/13.37	24 h	Breast cancer/MCF-7	>30 μg/mL	[89]
*Citrullus colocynthis*	Seeds	Spherical/16.57	24 h	Breast cancer/MCF-7	>30 μg/mL	[89]
*Erythrina indica*	Root	Spherical/20–118	Overnight	Breast cancer/MCF-7	-	[98]
*Olax scandens*	Leaf	Spherical/30–60	2 h	Breast cancer/MCF-7	Dose dependent	[106]
*Piper longum*	Fruit	Spherical/46	24 h	Breast cancer/MCF-7	67 μg/mL	[107]
*Rheum emodi*	Root	Spherical/27.5	24 h	Breast cancer/MCF-7	Dose dependent	[108]
*Syzygium cumini*	Flower	Spherical/40	6 h	Breast cancer/MCF-7	Dose dependent	[82]
*Taxus baccata*	Needles	Spherical/56	10 min	Breast cancer/MCF-7	37 μg/mL	[109]
*Syzygium aromaticum*	Fruit	Spherical/5–20	20 min	Breast cancer/MCF-7	70 μg/mL	[96]
*Ulva lactuca*	Whole plant	Spherical/56	10 min	Breast cancer/MCF-7	37 μg/mL	[109]
*Achillea biebersteinii*	Flower	Spherical, pentagonal/12	3 h	Breast cancer/MCF-7	20 μg/mL	[110]
*Azadirachta indica*	Leaf	Spherical/<40	6 h	Breast cancer/MCF-7	Dose dependent	[82]
*Melia dubia*	Leaf	Irregular/7.3	15 min	Breast cancer/MCF-7	31.2 μg/mL	[111]
*Sesbania grandiflora*	Leaf	Spherical/22	24 h	Breast cancer/MCF-7	20 μg/mL	[112]
*Citrullus colocynthi s*	Callus	Spherical/31	24 h	Laryngeal Cancer/Hep-2	3.42 μg/mL	[113]
*Suaeda monoica*	Leaf	Spherical/31	5 h	Laryngeal Cancer/Hep-2	500 nM, AgNPs conc.	[114]
*Ulva lactuca (algae)*	Whole plant	Spherical/56	10 min	Laryngeal Cancer/Hep-2	12.5 μg/mL	[109]
*Rubus glaucus Benth*	Root	Quasi-spherical/12–50	48 h	Hepatic cancer/Hep-G2	Dose dependent	[115]
*Citrullus colocynthis*	Root	Spherical/7.39	24 h	Hepatic cancer/Hep-G2	17.2 μg/mL	[116]
*Citrullus colocynthis*	Fruit	Spherical/19.26	24 h	Hepatic cancer/Hep-G2	22.4 μg/mL	[116]
*Citrullus colocynthis*	Leaf	Spherical/13.37	24 h	Hepatic cancer/Hep-G2	10.02 μg/mL	[116]
*Sargassum vulgare*	Whole plant	Spherical/10	3 h	Leukemia cancer/HL-60	Dose dependent	[79]
*Dimocarpus longan*	Peel	Spherical/8–22	2 h	Leukemia cancer/H1299	5.33 μg/mL	[117]
*Azadirachta indica*	Leaf	Spherical/< 40	6 h	Kidney cancer/Hek-293	Dose dependent	[82]

**Table 2 cancers-13-02818-t002:** List of studies exhibiting biosynthesized gold NPs and their anticancer activity.

Plant	Plant Part Used	Morphology/Size (nm)	Exposure Time	Cancer Type/Cell Line Used	IC_50_ Value	Ref.
*Azadirachta indica*	Leaf	Spherical, triangular, hexagonal	48 h	Cervical cancer/HeLa	No toxicity	[133]
*Genipa americana L.*	Fruit	Spherical/30.4 ± 14.9	48 h	Cervical cancer/HeLa	No toxicity	[134]
*Dracocephalum kotschyi*	Leaf	Spherical/11	24 h, 48 h, 72 h	Cervical cancer/HeLa	152.16 µg/mL	[135]
*Zataria multiflora*	Leaf	Pentagon, triangular/10–50	48 h	Cervical cancer/HeLa	100 µg/m	[132]
*Areca catechu*	Nut	Spherical/22.2	24 h	Cervical cancer/HeLa	25.17 µg/mL	[136]
*Mimosa pudica*	Leaf	Spherical/12	24 h, 48 h	Breast cancer/MCF-7	6 µg/mL	[121]
*Musa paradisiaca (banana)*	Stem	Spherical/30	24 h	Breast cancer/MCF-7	Low toxicity	[122]
*Antigonon letopus Hook. and Arn.*	Aerial part	Spherical, triangular/13–28	48 h	Breast cancer/MCF-7	257.8 μg/mL	[123]
*Corallina officinalis*	Aqueous Extract	Spherical/14.6	NA	Breast Cancer/MCF-7	NA	[125]
*Phoenix dactylifera*	flower	Near spherical/95	24 h	Breast Cancer/MCF-7	4.76 μg/mL	[126]
*Vites vinefera*	Aqueous Extract	Spherical/20–45	24 h	Breast Cancer/HBL- 100	NA	[127]
*Acalypha indica*	Leaf	Spherical/20–30	30 min	Breast Cancer/MDA- MB-231	NA	[124]
*Alternanthera bettzickiana*	Leaf	Spherical and aggregated/80–120	10 min	Lung Cancer/A549	NA	[128]
*Sesuvium portulacastrum*	Leaf	Mostly Spherical/35–40	0–8 h	Lung Cancer/A549	14 μg/mL	[129]
*Star anise (Illicium verum)*	Pod	Hexagonal, triangular/20–150	48 h	Lung cancer/A549	Low toxicity	[130]
*Star anise (Illicium verum)*	Pod	Hexagonal, triangular/20–50	48 h	Lung cancer/A549	Low toxicity at 200 nM	[130]
*Musa paradisiaca (banana)*	Stem	Spherical/30	24 h	Kidney cancer/HEK293	>80 nM	[122]
*Ficus religiosa*	Bark	Spherical/20–30	24 h	Kidney cancer/HEK 293	No toxicity	[138]
*Hibiscus sabdariffa*	Leaf, stem	Near spherical/10–60	48 h	Kidney cancer/HEK 293	2 ng/mL	[137]
*Couroupita guianensis*	Flower	Polydispersed, spherical, triangular, tetragonal/7–48	5 min	Leukaemia/HL-60	NA	[139]
*Cajanus cajan*	Seed coat	Spherical/9–41	24 h	Liver cancer/HepG2	6 µg/mL	[140]

**Table 3 cancers-13-02818-t003:** List of studies exhibiting biosynthesized zinc NPs and their anticancer activity.

Plant	Plant Part Used	Morphology/Size (nm)	Exposure Time	Cancer Type/Cell Line	IC_50_ Value	Ref.
*Abutilon indicum*	Leaf	Spherical/35.2 ± 2.3	2–3 h	Lung cancer/Calu-6	9.34 ± 0.4 μg/mL	[143]
*Calotropis gigantea*	Leaf	Spherical/30–35	3 h	Lung cancer/Calu-6	11.6 ± 0.9 μg/mL	[144]
*Laurus nobilis*	Leaf	Hexagonal/47.27	4 h	Lung cancer/A549	11.3 ± 0.9 μg/mL	[142]
*Cannabis sativa*	Leaf	Hexagonal/40 ± 1.5	3 h	Lung cancer/A549	18.3 ± 1.3 μg/mL	[145]
*Calotropis procera*	Leaf	Spherical/5–40	4 h	Lung cancer/A549	15.2 ± 1.6 μg/mL	[146]
*Withania Somnifera*	Leaf	Hexagonal/51.34	2–3 h	Leukemia/WEHI-3	12.4 ± 1.6 μg/mL	[147]
*Sargassum muticum*	Leaf	Spherical/22.5 ± 3.5	3–4 h	Leukemia/WEHI-3	2.25 ± 0.4 μg/mL	[148]
*Tabernaemontana divaricate*	Leaf	Spherical/36 ± 5	3 h	Breast cancer/MCF-7	30.6 μg/mL	[149]
*Tabernaemontana divaricate*	Leaf	Spherical/36 ± 5	4 h	Breast cancer/MCF-7	30.6 μg/mL	[150]
*Tabernaemontana*	Leaf	Spherical/36 ± 5	3–4 h	Breast cancer/MCF-7	30 μg/mL	[151]
*Borassus flabellifer*	Leaf	Spherical/55	3 h	Breast cancer/MCF-7	0.125 μg/mL	[152]
*Embelia ribes*	Root	Spherical/130–150	2 h	Breast cancer/MCF-7	9.62 ± 1.9 μg/mL	[153]
*Saccharum officinarum*	Juice	Spherical/19 ± 2.3	4 h	Breast cancer/MCF-7	16.7 ± 0.5 μg/mL	[106]
*Anabaena variabilis*	Phyco-bili pigment	Spherical/42 ± 3	5–6 h	Breast cancer/MCF-7	16.5 1.6 μg/mL	[154]
*Atropa belladonna*	Leaf	Hexagonal/34 ± 3.2	2 h	Breast cancer/MCF-7	12 ±0.9 μg/mL	[160]

**Table 4 cancers-13-02818-t004:** List of studies exhibiting biosynthesized copper/copper oxide NPs and their anticancer activity.

Plant	Plant Part Used	Morphology/Size (nm)	Exposure Time	CancerType/Cell Line	IC50 Value	Ref.
*Azadirachta indica*	Leaf	Spherical/12	1 h	Cervical Cancer/HeLa	0.89 μg/mL	[164]
*Phaseolus vulgaris*	Seed	Spherical/26.6	7–8 h	Cervical Cancer/HeLa	NA	[163]
*Calotropis procera L.*	Latex	Spherical/5–30	24 h	Cervical Cancer/HeLa	No toxicity	[168]
*Azadirachta indica*	Leaf	Spherical/12	1 h	Breast cancer/MCF-7	27.4, 45.3, 37μg/mL	[164]
*Olea europaea*	-	Spherical/20–50	24 h	Breast cancer/AMJ-13	1.47 μg/mL	[166]
*Acalypha indica*	Leaf	Spherical/26–30	48 h	Breast cancer/MCF-7	56.16 μg/mL	[165]
*Ficus religiosa*	Leaf	Spherical/577	24 h	Lung cancer/A549	200 μg/mL	[167]
*Calotropis procera L.*	Latex	Spherical/55	24 h	Lung cancer/A549	No toxicity	[168]
*Azadirachta indica*	Leaf	Spherical//12	1 h	Lung cancer/A549	26.7, 21.6,μg/mL	[164]
*Olea europaea*	-	Spherical/20–50	24 h	Ovarian cancer/SKOV-3	2.27 μg/mL	[166]
*Broccoli*	Whole plant	Spherical/∼4.8	2 h	prostate cancer/PC-3	No toxicity	[169]

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
