# Peer review of "A Systematic Review of Biosynthesized Metallic Nanoparticles as a Promising Anti-Cancer-Strategy"

_cancers, 2021, doi:10.3390/cancers13112818_

Round 1

Reviewer 1 Report

This is a very interesting and promising paper which is certainly usefull because it concerns the application of green nanoparticles (Nps) in tumor cell lines. The tabular analysis carried out on the different Nps is remarkable, although in my opinion the authors should better characterize the effect of the single plant extract to provide more information on the substances to be used combined with the different shapes and sizes of the Nps. Certainly, the road to the use of Nps inthe treatment of diseases is promising but still very tortuous.

Author Response

Dear Professor Ana Zubić, Dear Reviewer 1,

We would like to thank you and the reviewers of our manuscript for your constructive feedback. We are pleased to resubmit the revised version of our manuscript titled “A Systematic Review of Green Biosynthesized Metallic Nanoparticles as a Promising Anti-Cancer-Strategy” along with point-by-point response to the reviewer comments for publication in your prestigious journal; Cancers.

In this revision, we have carefully addressed all the comments raised by the reviewers. Please find below a detailed response to the reviewer comments, and a listing of all modifications made.

We look forward to the acceptance of this revised version for publication in Cancers.

Yours sincerely,

Dr Bilal Haider Abbasi, Dr Christophe Hano

Reviewer 1

Reviewer comment:

This is a very interesting and promising paper which is certainly usefull because it concerns the application of green nanoparticles (Nps) in tumor cell lines. The tabular analysis carried out on the different Nps is remarkable, although in my opinion the authors should better characterize the effect of the single plant extract to provide more information on the substances to be used combined with the different shapes and sizes of the Nps. Certainly, the road to the use of Nps inthe treatment of diseases is promising but still very tortuous.

Author’s Response:

We are grateful to Reviewer 1 for the critical reading of the manuscript and supportive comments to improve the quality and clarification. We sincerely thank the reviewer for his/her thoughtful suggestions. We agree that it would be more interesting to include the effect of the single plant extract to provide more information on the substances to be used combined with the different shapes and sizes of the Nps in this manuscript. But in order to keep this review article less verbose and to include more related literature, we have aimed to discuss biosynthesis and anticancer effects of different NPs from various plant species in a generalized way.

Reviewer 2 Report

This review gives an overview of the recent advancements in the biosynthesis of metallic nanoparticles (Ag, Au, Zn and Cu) and their possible anti-cancer activities, with particular focus on their mechanism of action. Nanoparticles mediated toxicity in cancerous cells, as well as the future prospects of nano-therapeutics are also discussed. This article provides an insight into the green synthesis of metallic nanoparticles and their potential applications as therapeutics in cancer therapy. This review has mainly focused on biosynthesis of silver, gold, zinc and copper nanoparticles for cancer therapy and their in vitro anticancer activities against cell lines. I would like to recommend this article for publication. However, I would like to mention few comments. They should be treated as minor:

  • In the abstract: take care of , (line 45)
  • Take care of spaces between words and/or symbols/references in the whole text:
  • line 167, 200, 348, 349
  • use italic form for in-vitro (line 386)
  • Change:
    • Line 316: IC50→IC50
    • Line 376: Zno→ZnO
    • Line 405: Cuo→CuO
    • Line 442: -NH2→-NH2

Author Response

Dear Professor Ana Zubić, Dear Reviewer 2,

We would like to thank you and the reviewers of our manuscript for your constructive feedback. We are pleased to resubmit the revised version of our manuscript titled “A Systematic Review of Green Biosynthesized Metallic Nanoparticles as a Promising Anti-Cancer-Strategy” along with point-by-point response to the reviewer comments for publication in your prestigious journal; Cancers.

In this revision, we have carefully addressed all the comments raised by the reviewers. Please find below a detailed response to the reviewer comments, and a listing of all modifications made.

We look forward to the acceptance of this revised version for publication in Cancers.

Yours sincerely,

Dr Bilal Haider Abbasi, Dr Christophe Hano

 Responses to Reviewer 2 Comments

Reviewer comment:

This review gives an overview of the recent advancements in the biosynthesis of metallic nanoparticles (Ag, Au, Zn and Cu) and their possible anti-cancer activities, with particular focus on their mechanism of action. Nanoparticles mediated toxicity in cancerous cells, as well as the future prospects of nano-therapeutics are also discussed. This article provides an insight into the green synthesis of metallic nanoparticles and their potential applications as therapeutics in cancer therapy. This review has mainly focused on biosynthesis of silver, gold, zinc and copper nanoparticles for cancer therapy and their in vitro anticancer activities against cell lines. I would like to recommend this article for publication. However, I would like to mention few comments. They should be treated as minor:

Author’s Response:

We would like to thank Reviewer 2 for the valuable comments on our manuscript. As per reviewer’s suggestions we have addressed all comments raised by the reviewer which can be seen highlighted in yellow background in the revised manuscript.

Reviewer comment#1: In the abstract: take care of, (line 45):

Author’s Response:

We would like to thank Reviewer 2 for this comment. We have revised the text of line 45 (highlighted in yellow) in the abstract of the revised draft, as:  

From: among these silver (Ag), gold (Au) zinc (Zn) and copper (Cu) are the leading anti-cancer NPs.

To: among these silver (Ag), gold (Au), zinc (Zn) and copper (Cu) are the leading anti-cancer NPs.

Reviewer comment#2: Take care of spaces between words and/or symbols/references in the whole text: line 167, 200, 348, 349.

Author’s Response:

We would like thank Reviewer 2 for this comment. We have revised the text (highlighted in yellow) in the revised draft by adding required spaces between words, symbols, and references in the suggested lines, as: 

 line 167:

From: having pharmacokinetics issues. [46, 47]. NPs are progressing as an attractive tool of research.

To: having pharmacokinetics issues [46, 47]. NPs are progressing as an attractive tool of research.

line 200:

From: enhanced blood vasculature due to upregulated expression of angiogenic factors [57]. [58].

To: enhanced blood vasculature due to upregulated expression of angiogenic factors [57, 58].

line 348:

From: 22–30nm showed cytotoxicity against MDA- MB-231 breast cancer cell lines by activating.

To: 22–30 nm showed cytotoxicity against MDA- MB-231 breast cancer cell lines by activating.

line 349:

From: apoptotic cell death pathways [124]. Bio-extract derived AuNPs with 14.6nm size exhibited.

To: apoptotic cell death pathways [124]. Bio-extract derived AuNPs with 14.6 nm size exhibited.

 Reviewer comment#3: use italic form for in-vitro (line 386)

Author’s Response:

We would like to thank Reviewer 2 for this comment. We have italicized and highlighted the term in-vitro in line 386 in the revised draft, as:

ZnO NPs and their anticancer effects have been investigated in-vitro using various cancerous.

 Reviewer comment#4: Change:

    • Line 316: IC50→IC50
    • Line 376: Zno→ZnO
    • Line 405: Cuo→CuO
    • Line 442: -NH2→-NH2

Author’s Response:

We thank Reviewer 2 for bringing this to our attention. We have made suggested changes in the recommended lines, as.

Line 316: IC50→IC50

laryngeal carcinoma cell line Hep-2, with IC50 values ranged between 3.42 μg/mL-12.5.

Line 376: Zno→ZnO

Applications of Biosynthesized Zinc and Zinc Oxide NPs (Zn/ZnO-NPs) as Anti-cancer.

Line 405: Cuo→CuO

Applications of Biosynthesized Copper/Copper Oxide NPs (Cu/CuO-NPs) as Anti-Cancer.

Line 442: -NH2→-NH2

are considered less toxic than –OH group and –NH2 groups [172]. Toxicity can also be.
